# The Design and Validation of a Compassion Fatigue Scale in Peruvian Nurses

**DOI:** 10.3390/ijerph20126147

**Published:** 2023-06-16

**Authors:** Renzo Felipe Carranza Esteban, Oscar Mamani-Benito, Josué E. Turpo Chaparro, Janett V. Chávez Sosa, Susana K. Lingan

**Affiliations:** 1Grupo de Investigación Avances en Investigación Psicológica, Facultad de Ciencias de la Salud, Universidad San Ignacio de Loyola, Lima 15088, Peru; 2Facultad de Derecho y Humanidades, Universidad Señor Sipán, Chiclayo 14000, Peru; 3Escuela de Posgrado, Universidad Peruana Unión, Lima 15457, Peru; 4Escuela Profesional de Enfermería, Universidad Peruana Unión, Lima 15457, Peru

**Keywords:** compassion fatigue, factor analysis, statistical, nurses, Peru, validation study

## Abstract

The objective of this study was to design and validate the Compassion Fatigue Scale (EFat-Com) in Peruvian nurses. Methods: A 13-item scale was designed using qualitative procedures and expert judgment. This version was administered to 201 nursing professionals using an electronic form along with two other measures: the Patient Health Questionnarie-2 and the Satisfaction with Life Scale. Results: Exploratory factor analysis supported the existence of two factors with factor loadings > 0.54. Confirmatory factor analysis of the two-factor model yielded satisfactory fit indices after the elimination of two items. Regarding concurrent validity, a positive relationship was obtained between the EFat-Com and the measure of depression; however, no correlation was found with the measure of life satisfaction. The internal consistency was 0.807 for the total scale, 0.79 for Factor 1, and 0.83 for Factor 2. Conclusions: The EFat-Com showed adequate psychometric properties with respect to content-based validity evidence, internal structure, and reliability. Therefore, the instrument can be used in research and professional settings. However, it is essential to continue studying the validity evidence in other contexts.

## 1. Introduction

In the context of health sciences, promoting well-being in the workplace is vital for health professionals to perform their work effectively [1]. This is all the more important in the context of the COVID-19 health emergency, wherein the nursing population has been exposed to stressful and traumatic situations [2].

The term *Fatiga por compasión* is a literal translation of the term compassion fatigue used by English-speaking researchers. Its origin dates back to the 1980s and early 1990s, when researchers’ attention shifted from burnout syndrome to the phenomenon that exposed the negative impact of working with people who had experienced traumatic events. Therefore, in 1992, Joinson [3] coined the term when he realized that healthcare workers who dealt with traumatized clients or patients could experience the same side effects.

From then, the term evolved and is currently defined as a syndrome entailing the symptoms of secondary trauma and burnout syndrome combined, in other words, an emotional response to events involving the suffering of a patient, vulnerable people, and/or people who have experienced some kind of trauma, which causes an alteration in the mental health of the healthcare professional who is suffering from burnout and professional exhaustion [4]. In this regard, one of the theoretical models that can best explain this phenomenon is the one proposed by Figley [5]. They argued that in order to experience compassion fatigue, some factors must be present, such as the professional’s capacity and empathic attitude, together with direct exposure to the pain narrative described by the patient. In this sense, this theoretical model is based on the assumption that prolonged exposure to suffering, including traumatic memories that the professional might have suffered, can cause compassion stress and result in the phenomenon of compassion fatigue.

A review of the available scientific literature shows that compassion fatigue is detrimental to the mental health and quality of life of healthcare professionals, especially nurses [6,7,8,9,10]. In the present context, this problem has been aggravated by the COVID-19 pandemic, especially during the first wave. Due to the work overload experienced during the health emergency, health workers reported psychological symptoms, post-traumatic stress, and compassion fatigue [11]. These issues were particularly experienced by those who worked in emergency departments and specific direct care units with COVID-19 patients [12]. In addition, the results of research on health professionals suggest that one of the populations at a high risk of developing compassion fatigue was the nurses working on the frontline [13] because they were exposed to the prolonged suffering of patients arriving at intensive care and emergency units [14].

Given the prevalence of this phenomenon, the scientific literature has illuminated the need for assessing the magnitude of this problem, thus different proposals arose for instruments aimed at assessing compassion fatigue. In this case, one of the first instruments was the 40-item Compassion Fatigue Self-Test (CFST) created by Figley [15], which also had a shorter version, *Compassion Fatigue-Short Scale* [16]. Subsequently, Stam and Figley [17] added items on positive aspects, parallel to the negative aspects, resulting in a test containing 66 items. Lastly, Gentry et al. [18] reported a different version of the CFST, which they called CFS-R, made up of 30 items. On the other hand, other authors developed alternative measures, such as the *Compassion Fatigue-Short Scale* by Adams et al. [19], which consists of 13 items, and in another scenario, the *Professional Quality of Life Scale (ProQOL)* [20,21], which includes three subscales and addresses compassion satisfaction, burnout, and compassion fatigue. The latter has been widely accepted in the Latin American scientific community, as evidenced in various validation studies [22,23,24,25,26,27].

In this case, there are two reasons behind the authors’ decision to design a new instrument instead of merely adapting an existing one. The first reason has to do with being aware that no measurement instrument can assess all aspects within the concept of compassion fatigue (among others: trauma symptoms, cognitive distortions, general psychological distress, and burnout) [28]. The second one is that the evaluation instruments would not be totally and directly adaptable to the Spanish-speaking population in Peru due to the existing differences in the type of professional practices and in the organization they entail. Therefore, the design of evaluation methods should consider cultural, social, and professional particularities [29].

In sum, nurses constitute a population that is vulnerable to psychological alterations that could affect their quality of life, mainly due to depression and compassion fatigue [27]. However, there is a lack of valid and reliable instruments in the Peruvian context, especially considering the recent events in the context of the COVID-19 pandemic, wherein it has been observed that the health system of this country has many limitations when it comes to protecting the mental health of its health workers [30]. They are exposed to long working hours during which they must deal with the trauma and suffering caused by the repercussions of SARS-CoV-2 [31].

The objective of this study was to design and validate the Compassion Fatigue Scale (EFat-Com) in Peruvian nurses.

## 2. Methodology

### 2.1. Design and Population

This study was designed as an instrumental and cross-sectional study [32].

The study was based on a nonprobabilistic sampling by volunteers, which included Peruvian nursing professionals who performed healthcare functions in private or state health institutions. The sample comprised 201 participants (85.6% women and 14.4% men) with an average age of 40.89 years (standard deviation = 10.24). Furthermore, 47.3% worked in critical care units (24.9% intensive care unit and 22.4% emergency), and 66.2% were employed (contracted full-time and contracted part-time).

### 2.2. Instrument

To elaborate the instrument, a literature review was conducted in the Scielo and Scopus databases for the theoretical framework to be built and the construct to be conceptually defined [10,16,33]. Based on this approach, the Compassion Fatigue in Health Care Personnel scale was designed.

Subsequently, the content validity evidence was analyzed to evaluate the relevance, representativeness, and clarity of the items, consisting of a group of four expert physicians and three psychologists who practiced in the hospital setting. Initially, the scale was composed of 13 items distributed across two factors (compassion and fatigue–exhaustion), whose response options are in Likert format, where 0 = never, 1 = rarely, 2 = sometimes, 3 = frequently, and 4 = always.

To analyze the validity evidence based on the relationship with other variables, the Patient Health Questionnarie-2 (PHQ-2) [34], a brief measure that analyzes depression and is composed of two items with four Likert-type response alternatives (from 0 (not at all) to 3 (almost every day)), was used. In this study, the PHQ-2 reported adequate reliability (α = 0.80, 95% confidence interval (CI): 0.74–0.84). In a similar way, the Satisfaction with Life Scale (SWLS; Diener et al., 1985) [35], which was validated in the Peruvian context by Caycho-Rodríguez et al., was used [36]. The scale is a brief instrument comprising five statements whose response options range from 1 (totally disagree) to 5 (totally agree). The reliability of the SWLS in this investigation was α = 0.90 (95% CI: 0.87–0.92).

### 2.3. Procedure

Information was collected using Google Forms (available between the months of February and August 2021) distributed via social networks (Facebook, Messenger, and WhatsApp); thus, health workers employed in national or private hospitals were contacted and asked to voluntarily participate. They were provided with a commitment to respect their privacy. Information on the purposes of the research was provided, and informed consent was obtained.

### 2.4. Data Analysis

Evidence of content validity was analyzed using Aiken’s V coefficient [37]. Descriptive analysis and exploratory factor analysis (EFA) were performed with the Factor Analysis Program, version 10.1. Furthermore, the mean, standard deviation, skewness, and kurtosis were analyzed for the 14 items of the scale by considering a skewness and kurtosis coefficient of −2 [38] and the Kaiser–Meyer–Olkin coefficient (KMO) and Bartlett’s test for EFA. Consequently, parallel analysis suggested the existence of two factors, and the estimation method was unweighted least squares with prominent rotation.

For the confirmatory factor analysis (CFA), the RStudio program was used. Structural equation modeling was considered, and the comparative fit index (CFI) and the Tucker–Lewis Index (TLI) were analyzed. Additionally, the parameters for the root mean square error of approximation (RMSEA) and the root mean square error rate were taken into account as per the criteria proposed by Hu and Bentler [39], who stated that the TLI and CFI should be >0.9 and the RMSEA should be <0.08. Finally, concurrent validity was analyzed using Pearson’s correlation coefficient, and the reliability of the scale was calculated using the SPSS version 25.0 statistical program and its respective confidence intervals [40].

### 2.5. Ethical Considerations

The research was approved by the ethics committee of the University of Peru (reference no. 2021-CEUPeU-0037).

## 3. Results

Based on the judgment of seven experts and using Aiken’s V, the relevance, representativeness, and clarity of the EFat-Com items were analyzed. Table 1 shows that all items received a favorable evaluation by the experts (V > 0, 70). Items 3, 7, and 12 were found to be the most important ones (V = 1.00; 95% CI: 0.85–1.00), items 7 and 12 the most representative (V = 1.00; 95% CI: 0.85–1.00), and items 3 and 7 the most understandable (V = 1.00; 95% CI: 0.85–1.00). Likewise, the values of the lower limits of the 95% CI were adequate, and all the values of the V coefficient were statistically significant.

### 3.1. Preliminary Item Analysis

Table 2 shows the descriptive statistics (mean, standard deviation, skewness, and kurtosis) of the 13 items of the EFat-Com.

### 3.2. Exploratory Factor Analysis

An EFA was performed after review of the KMO index (0.800) and Bartlett’s test (1213.7; gl = 78; *p* = 0.000), both of which were good. The unweighted least squares method with prominent oblique rotation was used, and parallel analysis was employed for factor determination, which revealed that there were two factors underlying the 13 items. The rotated solutions of the 13 items explained 54.10% of the total variance. Factor 1 (compassion) explained 31.36% of the variance, and Factor 2 (fatigue–exhaustion) explained 22.74%. All items presented saturations > 0.54 (Table 3).

### 3.3. Confirmatory Factor Analysis

Table 4 shows the CFA, which was used to verify the evidence of validity. Based on the internal structure of the EFat-Com, the results of the original model showed that the goodness-of-fit indices were deficient. Therefore, items 6 and 12 were eliminated using the index modification technique, and a satisfactory factor structure model was obtained. The fit indices showed that the proposed model was adequate (χ^2^ = 97.899, degrees of freedom (df) = 43, *p* = 0.000; CFI = 0.974; TLI = 0.966; RMSEA = 0.079; and SRMR = 0.071). In summary, the model of 11 items distributed across two factors was satisfactory.

### 3.4. Validity Based on the Relationship with Other Variables

Table 5 shows the calculation of correlation coefficients among EFat-Com, PHQ-2, and SWLS. It was found that EFat-Com was directly and statistically significantly related to PHQ-2 (r = 0.250, *p* < 0.01) and had a small effect size. However, there was no statistically significant and practical correlation between EFat-Com and SWLS.

### 3.5. Reliability

The reliability of the scale was estimated with Cronbach’s α coefficient. An acceptable value of 0.85 was obtained for the general scale (α = 0.807; 95% CI = −0.75); likewise, 0.83 for Factor 1 (α = 0.790; 95% CI = −0.73) and Factor 2 (α = 0.835; 95% CI = 0.78–0.87) were obtained, thereby showing that the scale scores were reliable.

## 4. Discussion

In clinical practice, empathy is a topic of growing research interest [41] because it is considered a central aspect of patient care [5,42]. Empathy may contribute to the development of compassion fatigue [38,39], which is a concept associated with burnout resulting from prolonged exposure to compassion stress among those working in a service-related profession [43].

Nonetheless, there are few psychometric studies that analyze and measure this construct [33,44], even more so when considering measurement instruments created or adapted for Latin America [4,22]. In the Peruvian case, the recent pandemic placed this country among the hardest hit in Latin America [31,45]. Therefore, this research aimed to design and validate EFat-Com in Peruvian nurses.

The evidence of an internal structure provided by the EFA supported the existence of two factors with acceptable factor loadings since the 13 items showed adequate values [46]. The a priori two-factor model, obtained by means of the CFA, showed poor goodness of fit. Hence, in the modification, two items were eliminated (6 and 12) and a satisfactory structural model was reached, with Factor 1 being compassion and Factor 2 being fatigue–exhaustion. On the other hand, the EFat-Com was based on logical and empirical analyses, which demonstrated the degree to which the relationship of the items and their factors fit the construct investigated.

The two-factor model proposed by this research appears to be congruent with each other, given that the capacity for compassion is something that is required for compassion fatigue to occur [8]. It is precisely this capacity for compassion that is viewed positively by nurses and is in line with the charitable theory [47,48]. On the other hand, nurses who experienced fatigue or burnout demonstrated a distancing from patients whom they knew required emotional support [8,49]. This observation is in agreement with Figley’s theory [5], which states that nurses experience burnout as well as physiological and emotional dysfunction as a result of prolonged exposure to compassion stress. It is also important to mention that no single measure of compassion fatigue assesses all aspects of the construct concept. It is likely that those connected to the healthcare field will need to use more than one measure to provide a complete picture of each person’s experience of compassion fatigue [28].

Regarding the evidence of validity based on the relationship with other variables, the EFat-Com scores correlated significantly and positively with the measure of depression, which is consistent with studies [50] conducted previously in samples of nurses. However, no significant relationship was found with the measure of life satisfaction, which may be explained by the use of a brief, general measure of life satisfaction, rather than a measure that assessed specific domains, such as job satisfaction.

On the other hand, the 11-item EFat-Com, being brief, allowed its effective application to the target population. The items assessed the burnout, frustration, and emotional exhaustion of nurses in their patient care. In this regard, a study by Labrague et al. [13] showed that nurses on the frontline in the battle against COVID-19 were highly vulnerable to compassion fatigue. Furthermore, Alharbi et al. [7] reported that work environment, age, and years of experience were the predictors of compassion fatigue.

Previous studies have come up with a few scales that are comparable to the one developed in the present research. The study by Sun et al. [16] bears a resemblance to the objectives of this research. However, in contrast to our scale, it converged with two factors with an ɑ of 0.95, showing excellent reliability. Furthermore, the study by Yildirim and Cavcav [28], similar to that of Sun et al., reported two factors with an ɑ of 0.90, showing very good reliability. On the other hand, the ProQOL instrument evaluated by Carvalho and Sá [23] converged on eight factors with an ɑ of 0.86 for the eight factors. In subsequent psychometric studies, the instrument was reported to have poor consistency [18], but it was reported to possess good consistency indices in the Brazilian context [19]. Likewise, the instrument developed by Eng et al. [41] converged on three factors with an ɑ of 0.90 for the first factor, 0.87 for the second factor, and 0.74 for the third factor, but this was given in a sample of psychologists. However, the values presented in the EFat-Com scale showed a reliable and reproducible scale in similar contexts.

With regard to internal consistency, the EFat-Com scale showed acceptable reliability by showing values of α = 0.807 [51]; this is proof that the instrument is reliable and responds to two factors. These results are expected because the scale has a brief format, where the standard value is under 0.90 [51]. Therefore, it can be stated that the EFat-Com scale shows favorable psychometric properties for measuring compassion fatigue in Peruvian nurses, although there are some advantages when compared to previously developed instruments. Among these is its factorial simplicity, which is consistent with recent theoretical evidence [5,8,49,50]. In addition, it is an easy and quick instrument to apply due to its brevity, compared to other measurement tools [17,18]. Lastly, it uses understandable language that can be utilized by different healthcare professionals in charge of patient care, which promotes the development of future research that may validate the instrument on other groups of healthcare professionals, and hence expand its functionality.

Among the limitations of the study is the type of sampling that probably evidences some type of bias. This bias would prevent the findings from being generalized to all contexts. We suggest that similar studies be conducted to confirm the findings, with the understanding that this construct requires continuous follow-up [15]. In addition, the data were collected at a single point in time; hence, it is important to develop reviews that examine the stability of the item scores via a test–retest process [51]. Finally, the application of this instrument is recommended in contexts wherein the nurses are faced with patient care and manifest various levels of anxiety or psychological distress.

## 5. Conclusions

In conclusion, the EFat-Com scale is a valid and reliable measure to assess the level of compassion fatigue in Peruvian nurses. The scale is a brief version, and because of its adequate psychometric properties based on content, internal structure, and reliability, it can also be used for other studies in the health context.

## Figures and Tables

**Table 1 ijerph-20-06147-t001:** Aiken’s V for the evaluation of the relevance, representativeness, and clarity of items of EFat-Com.

Items	Relevance (n = 7)				Representativeness (n = 7)				Clarity (n = 7)			
	M	SD	V	95% CI	M	SD	V	95% CI	M	SD	V	95% CI
Item 1	2.86	0.38	0.95	0.77–0.99	2.86	0.38	0.95	0.77–0.99	2.86	0.38	0.95	0.77–0.99
Item 2	2.86	0.38	0.95	0.77–0.99	2.86	0.38	0.95	0.77–0.99	2.86	0.38	0.95	0.77–0.99
Item 3	3.00	0.00	1.00	0.85–1.00	2.71	0.49	0.90	0.71–0.97	3.00	0.00	1.00	0.85–1.00
Item 4	2.71	0.49	0.90	0.71–0.97	2.57	0.53	0.86	0.65–0.95	2.71	0.49	0.90	0.71–0.97
Item 5	2.57	0.53	0.86	0.65–0.95	2.57	0.53	0.86	0.65–0.95	2.86	0.38	0.95	0.77–0.99
Item 6	2.57	0.53	0.86	0.65–0.95	2.86	0.38	0.95	0.77–0.99	2.71	0.49	0.90	0.71–0.97
Item 7	3.00	0.00	1.00	0.85–1.00	3.00	0.00	1.00	0.85–1.00	3.00	0.00	1.00	0.85–1.00
Item 8	2.86	0.38	0.95	0.77–0.99	2.86	0.38	0.95	0.77–0.99	2.86	0.38	0.95	0.77–0.99
Item 9	2.71	0.49	0.90	0.71–0.97	2.71	0.49	0.90	0.71–0.97	2.71	0.49	0.90	0.71–0.97
Item 10	2.57	0.53	0.86	0.65–0.95	2.71	0.49	0.90	0.71–0.97	2.71	0.49	0.90	0.71–0.97
Item 11	2.71	0.49	0.90	0.71–0.97	2.71	0.49	0.90	0.71–0.97	2.57	0.53	0.86	0.65–0.95
Item 12	3.00	0.00	1.00	0.85–1.00	3.00	0.00	1.00	0.85–1.00	3.00	0.00	0.86	0.85–1.00
Item 13	2.57	0.53	0.86	0.65–0.95	2.57	0.53	0.86	0.65–0.95	2.57	0.53	0.86	0.65–0.95

Note: M = mean, SD = standard deviation, V = Aiken’s V coefficient, 95% CI = Aiken’s V confidence interval.

**Table 2 ijerph-20-06147-t002:** Preliminary analysis of the EFat-Com scale items.

Variable	M	SD	A	K	h
Item 1	3.188	0.991	−0.040	−0.044	0.454
Item 2	2.513	1.031	0.236	−0.418	0.563
Item 3	2.991	0.974	−0.039	−0.112	0.387
Item 4	2.850	1.050	−0.009	−0.330	0.324
Item 5	2.120	1.076	0.609	−0.426	0.462
Item 6	1.897	0.909	0.889	0.361	0.636
Item 7	3.880	0.829	−0.543	0.130	0.422
Item 8	3.735	0.896	−0.386	−0.225	0.420
Item 9	4.056	0.832	−0.506	−0.473	0.609
Item 10	3.812	0.928	−0.486	−0.282	0.499
Item 11	4.303	0.861	−1.476	1.557	0.364
Item 12	4.551	0.727	−1.884	1.927	0.406
Item 13	4.423	0.754	−1.177	0.762	0.442

Note: M = mean; SD = standard deviation; As = coefficient of skewness; K = coefficient of kurtosis; h = commonality.

**Table 3 ijerph-20-06147-t003:** Exploratory factor analysis of the EFat-Com.

Items	F1	F2
1. Al término de la jornada laboral (turnos en el hospital) me siento agotada/o (at the end of the workday (hospital shifts), I feel exhausted).		0.652
2. Trabajar con pacientes me desgasta emocionalmente (working with patients wears me down emotionally).		0.751
3. Siento impotencia por no poder hacer más en la recuperación de mi paciente (I feel helpless for not being able to do more for my patient’s recovery).		0.598
4. Me siento frustrada/o cuando fallece uno de mis pacientes (I feel frustrated when one of my patients dies).		0.544
5. Me desmotiva no poder estar emocionalmente bien y hacer más en la recuperación de mi paciente (I am discouraged that I cannot be emotionally well and do more for the recovery of my patients).		0.677
6. Me siento desgastado emocionalmente para poder atender oportunamente a mis pacientes (I feel emotionally drained while providing timely care for my patients).		0.803
7. Comprendo los sentimientos de frustración de los pacientes y/o familiares (I understand the frustration of patients and/or family members).	0.635	
8. Puedo ponerme en el lugar de los pacientes y/o familiares y sentir el dolor que ellos sienten (I can put myself in the place of patients and/or family members and feel the pain they feel).	0.610	
9. Siento compasión por los pacientes cuando pasan por dolor o sufrimiento (I feel compassion for patients when they endure pain or suffering).	0.782	
10. Soy sensible al dolor que experimentan los pacientes internados (I am sensitive to the pain experienced by hospitalized patients).	0.684	
11. Cuando sé cómo se siente el paciente en su proceso de recuperación, estoy más dispuesta(o) a ayudarlo (when I know how the patient feels in the recovery process, I am more willing to help him/her).	0.607	
12. Ayudar a los pacientes me hace sentir bien conmigo misma/o (helping the patients makes me feel good about myself).	0.641	
13. Cuando algún paciente está en riesgo de morir, siento compasión y trato de aliviar su dolor y/o sufrimiento (when a patient is at risk of dying, I feel compassion and try to alleviate their pain and/or suffering).	0.671	

Note: F1: compassion; F2: fatigue–exhaustion.

**Table 4 ijerph-20-06147-t004:** Fit indices of the models evaluated by confirmatory factor analysis of the study instrument.

Model	h^2^	df	CFI	TLI	RMSEA		SRMR	WRMR
					Value	90% CI		
13 items	2230.271	64	0.942	0.929	0.112	[0.096, 0.128]	0.101	10.309
11 items	970.899	43	0.974	0.966	0.079	[0.059, 0.100]	0.071	0.926

Note: df = degree of freedom; CFI = comparative fit index; TLI = Tucker–Lewis index; RMSEA = root mean square error of approximation; CI = confidence interval.

**Table 5 ijerph-20-06147-t005:** Means, standard deviations, and correlations among the EFat-Com, PQH-2, and SWLS.

Variable	M	SD	1	2
1. EFat-Com	41.61	6.60		
2. PQH-2	0.77	1.33	0.250 **	
3. SWLS	15.41	2.87	−0.015	−0.366 **

Note: M = mean; SD = standard deviation; ** indicates *p* < 0.01; PHQ-2 = depression; SWLS = satisfaction with life.

## Data Availability

The data presented in this study are available on request from the corresponding author.

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
