# Peer review of "The Design and Validation of a Compassion Fatigue Scale in Peruvian Nurses"

_ijerph, 2023, doi:10.3390/ijerph20126147_

Round 1

Reviewer 1 Report (Previous Reviewer 2)

Based on my previous reports, I believe that the authors addressed my suggestions.

This manuscript is a resubmission of an earlier submission. The following is a list of the peer review reports and author responses from that submission.

Round 1

Reviewer 1 Report

Thank you very much for letting me review this manuscript, which I found many interesting points. However, still I got some concerns,

Research motivation

1.     Why we need to have a new scale? a new compassion fatigue scale? More thorough literature review should be conducted.

2. More detailed information and specific information about how you generate the items can be provided. 

3. I wonder the difference between this compassion fatigue scale and previous similar scales.

Author Response

Reviewer 1

Why we need to have a new scale? a new compassion fatigue scale? More thorough literature review should be conducted.

Answer: A new review of the existing literature on the availability of instruments to assess compassion fatigue has been carried out and key information has been included. Furthermore, a paragraph justifying the decision to design a new scale has been added.

  1. More detailed information and specific information about how you generate the items can be provided. . 

Answer: The literature review reveals that there is theoretical information on compassion fatigue; however, instruments in Spanish that evaluate this construct are limited, so a bibliographic review was carried out in the Scielo and Scopus databases to build up the theoretical framework and to conceptually define the construct. The proposals were as follows:

Borges EMDN, Fonseca CIN da S, Baptista PCP, Queirós CML, Baldonedo-Mosteiro M, Mosteiro-Diaz MP. Compassion fatigue in nurses from an adult emergency and emergency hospital service. Rev Lat Am Nursing [Internet]. 2019;27:1–6. Available from: http://www.scielo.br/scielo.php?script=sci_arttext&pid=S0104-11692019000100360&tlng=pt

Hagan JL. Psychometric Evaluation of the ProQOL Version 5 for Assessing Compassion Satisfaction, Burnout and Secondary Traumatic Stress in Nurses. Int J Stud Nurs [Internet]. 2019 Jun 21;4(3):60–70. Available from: http://journal.julypress.com/index.php/ijsn/article/view/620

  1. I wonder the difference between this compassion fatigue scale and previous similar scales.

Answer: The information requested has been added.

Reviewer 2 Report

In this article, the validity and reliability of a Compassion fatigue Scale in Peruvian nurses, have been analysed. The analysis based on the study carried out on 201 nurses.

The article has some drawbacks. First, the analysis of the instrument has been carried out on a rather small study group, that represented only one vocational group. So, there was no possibilities to test validity and reliability of this instrument in relation to wider spectrum of professions (other health professionals).

- In the Introduction section please correct the first reference (18?).

- Under Methods, the scale needs to be described in more detail. Do not assume that the readers know about the tool. If permitted by the owner, include it as an appendix.

- The respondents' characteristics does not represent a good distribution of the population. Justification is needed why the results are still valid.

- the authors stated that 66,2% of the sample were employed, which means that 33,8% were unemployed. This is a serious limitation, since you should include this as a exclusion criteria to avoid bias. How do you guarantee that those nurses have enough work experience to respond? Were they employed during the COVID-19 first wave?

- Also, the authors could present the scores obtained to evaluate the level of fatigue among nurses and comment those results.

Author Response

Reviewer 2

- In the Introduction section please correct the first reference (18?).

Answer: The reference has been fixed.

- Under Methods, the scale needs to be described in more detail. Do not assume that the readers know about the tool. If permitted by the owner, include it as an appendix.

Answer: Emphasis is placed on the fact that the scale consists of thirteen items distributed into two factors (Compassion and fatigue-exhaustion), whose response options are expressed in Likert format, where 0 = never, 1 = rarely, 2 = sometimes, 3 = frequently, and 4 = always.

Table 3 shows the mentioned items.

- The respondents' characteristics does not represent a good distribution of the population. Justification is needed why the results are still valid.

Answer: There are several professionals in the field of health sciences; however, nurses are considered to be a vulnerable population and at risk of suffering psychological alterations that can affect their quality of life, considering that they play a role in the care of patients and also provide information to the family members of patients. Therefore, in this particular occasion the study has not been extended to other professions.

The results are valid since the EFat-Com scale reported that the psychometric properties are suitable for measuring compassion fatigue in Peruvian nurses. Furthermore, there are some advantages in comparison with previously developed instruments. For example; its factorial simplicity is consistent with the existing theoretical evidence. It is also an instrument that is easy and quick to apply due to its brevity and lastly, it uses understandable language.

- the authors stated that 66,2% of the sample were employed, which means that 33,8% were unemployed. This is a serious limitation, since you should include this as a exclusion criteria to avoid bias. How do you guarantee that those nurses have enough work experience to respond? Were they employed during the COVID-19 first wave?

Answer: In Peru, there are different types of contract modalities. Indeed, there are full-time employees or contracted workers, but there are also part-time employees and some employees who receive their salary through receipts for fees, the latter accounting for 33.8% of the total number of employees. However, this does not mean that they do not have sufficient work experience; it means that they do not lie within the category of permanent employees.

- Also, the authors could present the scores obtained to evaluate the level of fatigue among nurses and comment those results.

Answer: We believe that, on this particular occasion, those scores are not reported, since the objective and design of the study intends to be an instrument proposal where the psychometric properties (validity and reliability) of the instrument are analyzed.

To evaluate the level, scales must be used, which are tools used to classify the levels presented by people when compared to a given group. The purpose is directed more toward individual or clinical care (Ventura, 2022).

Round 2

Reviewer 1 Report

Thank you for letting me review this manuscript again. I believe that most of current concerns have been well addressed.

Author Response

Thanks so much.

Blessings

Reviewer 2 Report

The authors addressed my suggestions, but I believe that should be included a note regarding contract modalities, because for me all participants should be considered as employed despite the "type of contract". 

Author Response

Dear Reviewer.

The information requested has been added.
